

# Genotypic richness predicts phenotypic variation in an endangered clonal plant

Suzanna M. Evans[1,2], Elizabeth A. Sinclair[3,4], Alistair G.B. Poore[2], Keryn F. Bain[2] and Adriana Vergés[1,2]

[1] Centre for Marine Bio-Innovation, University of New South Wales, Sydney, New South Wales, Australia
[2] Evolution & Ecology Research Centre, University of New South Wales, Sydney, New South Wales, Australia
[3] School of Plant Biology and Oceans Institute, University of Western Australia, Perth, Western Australia, Australia
[4] Science Directorate, Botanic Gardens and Parks Authority, West Perth Western Australia, Australia

Corresponding author
Suzanna M. Evans,
s.evans@unsw.edu.au

## ABSTRACT

Declines in genetic diversity within a species can affect the stability and functioning of populations. The conservation of genetic diversity is thus a priority, especially for threatened or endangered species. The importance of genetic variation, however, is dependent on the degree to which it translates into phenotypic variation for traits that affect individual performance and ecological processes. This is especially important for predominantly clonal species, as no single clone is likely to maximise all aspects of performance. Here we show that intraspecific genotypic diversity as measured using microsatellites is a strong predictor of phenotypic variation in morphological traits and shoot productivity of the threatened, predominantly clonal seagrass *Posidonia australis*, on the east coast of Australia. Biomass and surface area variation was most strongly predicted by genotypic richness, while variation in leaf chemistry (phenolics and nitrogen) was unrelated to genotypic richness. Genotypic richness did not predict tissue loss to herbivores or epiphyte load, however we did find that increased herbivore damage was positively correlated with allelic richness. Although there was no clear relationship between higher primary productivity and genotypic richness, variation in shoot productivity within a meadow was significantly greater in more genotypically diverse meadows. The proportion of phenotypic variation explained by environmental conditions varied among different genotypes, and there was generally no variation in phenotypic traits among genotypes present in the same meadows. Our results show that genotypic richness as measured through the use of presumably neutral DNA markers does covary with phenotypic variation in functionally relevant traits such as leaf morphology and shoot productivity. The remarkably long lifespan of individual *Posidonia* plants suggests that plasticity within genotypes has played an important role in the longevity of the species. However, the strong link between genotypic and phenotypic variation suggests that a range of genotypes is still the best case scenario for adaptation to and recovery from predicted environmental change.

## INTRODUCTION

The link between biodiversity and ecosystem function is well-established at the species level, with communities that support a wide range of species often better able to stabilise multiple ecosystem processes in response to disturbance or change than species poor communities (*Loreau et al., 2001*; *Hooper et al., 2005*; *Stachowicz, Bruno & Duffy, 2007*; *Tilman, Reich & Isbell, 2012*; *Cardinale et al., 2013*; *Lefcheck et al., 2015*). Similarly, genetic diversity within species, as measured by the number of genotypes, can enhance productivity (*Aguirre & Marshall, 2012*), increase resilience (*Massa et al., 2013*) and have cascading benefits to the surrounding ecosystem (*Hughes et al., 2008*; *Ellers, 2009*). The importance of genetic variation to ecosystem functioning, however, depends on the degree to which it translates into variation in traits that affect the functioning of individuals, and upon which natural selection can act (*Foster Huenneke, 1991*; *Vasemägi & Primmer, 2005*; *Bolnick et al., 2011*). Therefore, it is necessary to quantify the degree of phenotypic variation among genotypes within species to better predict how loss of genotypes can affect the performance of populations (*Lande & Shannon, 1996*; *Ellers, 2009*; *Forsman & Wennersten, 2015*), particularly during periods of changing climatic conditions and increased anthropogenic stressors (*Best, Stone & Stachowicz, 2015*).

The conservation of genetic diversity is a priority for the management of threatened or endangered species, with the aim of maintaining evolutionary viability by maximising the chances of persistence in the face of environmental change (*Foster Huenneke, 1991*; *Reed & Frankham, 2003*; *Hughes et al., 2008*). For species that have naturally low levels of genetic variation due to predominantly clonal life history strategies, differences in functioning among the few genotypes present becomes especially important. It may be that only the most diverse assemblage will be able to maximise multiple ecosystem functions (*Duffy, Richardson & Canuel, 2003*), which suggests that a loss in diversity may significantly influence ecosystem properties. The theory that higher species diversity benefits ecosystem 'multifunctionality' stems from the observation that species that do not contribute to one ecosystem process often play an important role in a separate process and/or under different conditions (*Bradford et al., 2014*; *Byrnes et al., 2014*; *Lefcheck et al., 2015*). Similarly, it is highly unlikely that any single genotypic clone, no matter how well-adapted, would be able to capitalise on all aspects of performance. Rather, it is expected that multiple genotypes will contribute to multiple different processes and/or thrive under different conditions.

While clonal species may be associated with lower genetic diversity, there are a number of benefits that result from clonal reproduction in plant species. For example, clones are able to translocate resources between connected ramets—potentially increasing the allocation of limiting resources to the tissues in which they are scarce, thus increasing the survival and distribution of the genet (*Alpert & Stuefer 1997*). Although this strategy may increase plant performance under many conditions, physical disturbance, fragmentation, and pathogens can cut off transport between ramets and potentially wipe out entire genets that are not adapted to these stressors (*Callaghan et al., 1992*; *Honnay & Bossuyt, 2005*). For populations in stable environments (i.e., biotic and/or abiotic conditions

remain constant), differences in the success of particular genotypes, as well as genetic drift, can lead to an overall decrease in genet number over time and the dominance of one or a few clones (*McLellan et al., 1997*; *Eckert, 2002*; *Balloux, Lehmann & De Meeûs, 2003*). In changing environments, however, persistence should be enhanced by the maintenance of genetic variation via occasional seed production or the recruitment of new sexually produced individuals (*Eckert, 2002*; *Honnay & Bossuyt, 2005*). Otherwise, the population is assumed to be at much greater risk of extinction if environmental conditions change, compared to a genetically diverse population of conspecifics. In this study, we examine the relationship between genetic diversity, variation in phenotype, and ecosystem processes in predominantly clonal, threatened seagrass meadows that range from near-monoclonal to genotypically diverse.

Seagrasses are a major source of primary productivity supporting food webs globally (*Mateo et al., 2006*; *Hughes, Stachowicz & Williams, 2009*), with an estimated value of over $US28, 000 ha$^{-1}$ year$^{-1}$ (adjusted for inflation; *Costanza et al., 1997*). Seagrass meadows also act as significant 'blue carbon' stores (*Fourqurean et al., 2012*) and have cascading benefits to surrounding ecosystems (*Duffy, 2006*; *Barbier et al., 2011*). Unfortunately, seagrasses are rapidly declining worldwide and are now rated amongst the most threatened ecosystems on the planet (*Waycott et al., 2009*). Over the past decade, manipulations of genotypic diversity and identity in *Zostera marina* seagrass meadows from the Northern Hemisphere have shown that increasing the number of genotypes within an experimental plot can enhance resistance to disturbance by grazers (*Hughes & Stachowicz, 2004*), influence grazer biomass (*Hughes, Best & Stachowicz, 2010*), and result in greater shoot densities and biomass compared to monocultures during disturbance events (*Reusch et al., 2005*; *Hughes & Stachowicz, 2011*). Experimental manipulations of clonal identity have shown wide variation in biomass production and herbivore susceptibility among individual genotypes that can be similar in magnitude to the variation caused by environmental factors (e.g., nitrogen loading; *Tomas et al., 2011*). The positive effects of higher genotypic richness can be attributed to a combination of (i) complementarity among individual genotypes, in which the environment is exploited more efficiently by including a wider variety of functionally important phenotypes, and (ii) the sampling effect, in which there is simply a statistically higher probability of selecting pre-adapted phenotypes from a more diverse group (*Hughes et al., 2008*; *Hughes & Stachowicz, 2011*; *Forsman & Wennersten, 2015*). However, despite the obvious link between genotypic diversity and the performance and functioning of seagrass meadows, there is a lack of information regarding the mechanisms by which this relationship occurs.

The majority of studies that have manipulated genotypic diversity in seagrass meadows have identified genotypes using allozymes or microsatellite DNA markers that are presumed to be selectively neutral (*Reusch, 2001*). As such, the ecological effects of these genotypic manipulations will depend on the relationship between these neutral markers and phenotypic variation (*Hughes et al., 2008*). Although it is frequently assumed that a greater diversity of phenotypic traits related to ecosystem function (e.g., variation in height or biomass) directly results from corresponding genetic diversity, this relationship is rarely directly tested (excluding some well-studied terrestrial examples from *Populus*
hybrid complexes, e.g., *Whitham et al., 2006*). However, in cases where the conservation of a species may be dependent on the ability to respond positively to stress and adapt to new and potentially challenging conditions, variation in both genotype and phenotype needs to be considered (*Hughes, 2014*).

On the east coast of Australia, meadows of the temperate seagrass species *Posidonia australis* Hook. *f* . vary widely in genetic diversity; from predominantly clonal to moderate genetic diversity (with sexual reproduction; *Evans et al., 2014*). As such, it provides an ideal model ecosystem to explore whether genetic diversity can predict variation in phenotypic and ecosystem processes. The rapid decline in *Posidonia australis* meadows associated with human population growth and urbanisation (*NSW Department of Primary Industries, 2012*; *West, 2012*) has increased the need to identify mechanisms behind variation in meadow form and function, so that we can better tailor our efforts to identify meadows at risk of extinction, as well as more successful avenues for restoration.

In this study, we quantified the relationship between genotypic richness and traits of *Posidonia australis* that are considered functionally important at the ecosystem level; plant productivity, leaf structure and chemistry, epiphyte load, and tissue loss to herbivory. Structural leaf traits (morphology, biomass and shoot density) provide a measure of habitat structure and complexity (*Middleton et al., 1984*). They also influence trophic interactions by providing shelter for both predators and prey (*Farina et al., 2009*) and are known to change in response to environmental stressors (*Waycott, Longstaff & Mellors, 2005*). Leaf chemical constituents such as nitrogen and phenols affect decomposition rates and photosynthetic capacity (*Harrison, 1989*; *Alcoverro, Manzanera & Romero, 2001*) and influence susceptibility to grazing (*Goecker, Heck & Valentine, 2005*; *Vergés et al., 2007*). Elevated epiphyte loads have been linked to human-induced eutrophication, can be potentially fatal to seagrasses by reducing light availability (*Borowitzka, Lavery & Van Keulen, 2006*), and are thus considered important indicators of ecosystem health (*Wood & Lavery, 2000*). Finally, tissue loss to herbivores is an important process that transfers energy to higher trophic levels and can influence the distribution and growth patterns of seagrasses (*Valentine & Duffy, 2006*). More specifically, our study addressed the following questions:

1. Does genotypic diversity in *P. australis* meadows predict phenotypic variation?
2. Are genotypically diverse meadows more productive?
3. Is genotypic diversity related to herbivory/epiphyte load?
4. What is the relative importance of genotypic identity and the environment in explaining phenotypic variation?

## MATERIALS AND METHODS

### Study species and location

*Posidonia australis* is a long-lived, slow-growing seagrass endemic to the temperate Australian coastline (*Gobert et al., 2006*). It is considered one of the most structurally complex seagrasses (*Middleton et al., 1984*) and provides food and refuge for a range of commercially important fish and invertebrate species (*Gobert et al., 2006*). *Posidonia*

*australis* is in rapid decline, particularly on the eastern coastline of Australia, where it has been formally listed as endangered in six estuaries (*NSW Department of Primary Industries, 2012*). In 2011, 360 individual shoots were collected from 12 geographically distinct meadows ($n = 30$ per meadow) across New South Wales (NSW) on the east coast of Australia, including the extreme northern range-edge of the species (Fig. 1). Shoots were collected once every 2–3 m along a linear transect, to reduce the chances of resampling the same genetic individual (*Evans et al., 2014*). Sample collections were supported by a NSW Department of Primary Industries Scientific Collection Permit (P11/0059-1.2).

## Genetic diversity

Immediately after collection, shoots were placed on ice and returned to the laboratory. A set of eight polymorphic microsatellite markers (developed by *Sinclair et al., 2009*) were used to determine levels of genetic diversity within and among the 12 *P. australis* meadows sampled. Following amplification (see *Evans et al., 2014* for full protocol), multiple PCR products were combined where possible (pre-determined by size and label) and run on a CEQ 8800 Genetic Analysis System (Beckman Coulter). Size standard 400 was used to determine allele sizes, which were then scored using the Beckman Coulter software. The number of unique multilocus genotypes (MLGs) and alleles per meadow were estimated using GenClone 2.0 (*Arnaud-Haond & Belkhir, 2007*), as well as the likelihood of individual genotypes arising through sexual recombination given allelic frequencies in the regional population ($P_{gen}$). Allelic richness ranged from 10 to 20 alleles per meadow, while genotypic richness ranged from as few as two MLGs per meadow (near-monoclonal) through to 21 MLGs per meadow (high diversity). In total, 79 unique MLGs were identified from the 360 shoots sampled. Of these, seven MLGs were shared across two or more meadows (Table 1), while the remainder were unique to their sample locations. For all genotypes $P_{gen}$ was <0.02, allowing us to conclude that individual shoots sharing the same genotype are highly unlikely to have arisen through sexual recombination (*Arnaud-Haond & Belkhir, 2007*).

## Phenotypic diversity

Shoots collected for genotyping were measured for traits relating directly to the shoot phenotype (leaf surface area, biomass, nitrogen content, and total phenolics), as well as traits considered important for ecosystem functioning (productivity, herbivory, and epiphyte biomass). In addition, shoot density and herbivorous fish counts were done within each meadow.

In the laboratory, epiphytes were carefully removed from the leaves with a razor blade. While some shoots retained small quantities of encrusting epiphytes on the oldest leaves, this did not impact morphometric measurements, and these leaves were not included in chemical analyses. Morphometric measurements were made on all leaves present (generally between 2–5 leaves per shoot), from the ligule to the leaf tip. Surface area of all leaves was calculated as $mm^2$ $shoot^{-1}$, minus any damage due to herbivory. The leaf biomass (mg $shoot^{-1}$) of each shoot was determined by removing the leaves at the ligule,

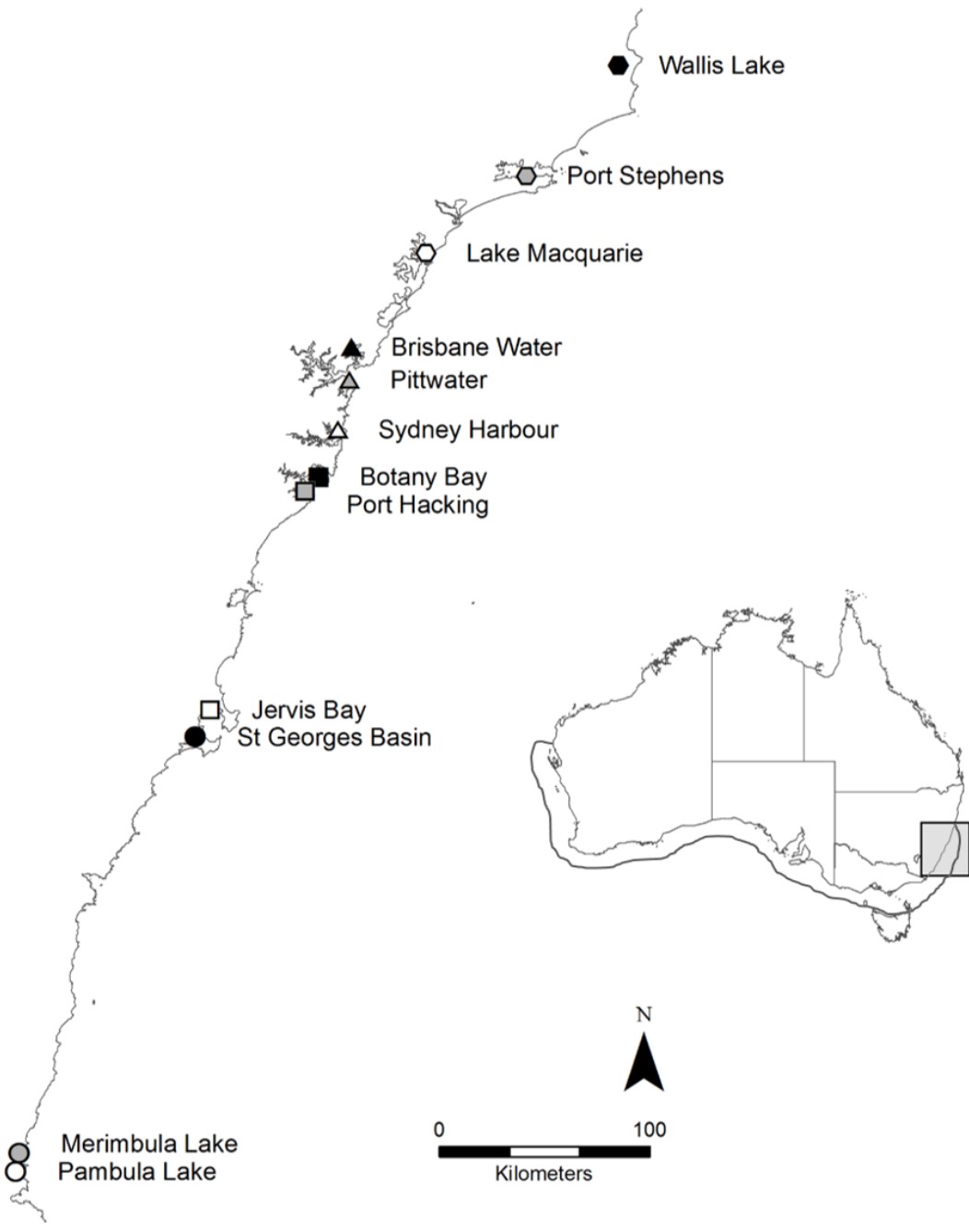

**Figure 1 Map of the twelve sample locations in New South Wales, Australia.** The full distribution of *P. australis* is represented by the outline on the inlaid map. The ∼600 km of coastline sampled on along the east coast is highlighted by the light grey box.

freeze-drying and then weighing. Any attached rhizome was not included as the amount collected varied widely between shoots. Meristematic tissue below the ligule was also excluded as this was removed for DNA analysis.

Shoot density was determined *in situ* using a 0.25 m$^2$ quadrat in which all individual shoots were counted. The quadrat was haphazardly distributed ten times within each
**Table 1** **Frequency of the seven shared multilocus genotypes in New South Wales meadows.** Frequency of the seven shared multilocus genotypes (MLGs) originally identified in *Evans et al. (2014)*. These seven MLGs are arbitrarily named using the letters A–G. The frequency with which they occur in each meadow is shown in the columns titled 'N per meadow'. The meadows in which these shared genotypes are found are listed. All remaining MLGs not listed were unique to their sample location. Four meadows not listed here contained only unique MLGs.

| MLG | Total N | N per meadow | | | | | | |
|---|---|---|---|---|---|---|---|---|
| | | Wallis Lake | Port Stephens | Lake Macquarie | Brisbane Water | Pittwater | Sydney Harbour | Botany Bay |
| A | 56 | 29 | | 27 | | | | |
| B | 2 | 1 | | 1 | | | | |
| C | 21 | | 13 | | | 8 | | |
| D | 11 | | 7 | | | 2 | | 2 |
| E | 4 | | | 2 | | | | 2 |
| F | 29 | | | | 28 | | 1 | |
| G | 21 | | | | | 20 | | 1 |

meadow to obtain a mean shoot density per meadow. Quadrat data were recorded at the same time and within the same meadow area as the genetic sampling. Productivity per shoot was measured with lepidochronology, which uses patterns in the thickness of leaf sheaths retained on the rhizome from old leaves to estimate the rate of new leaf production over time, recorded as dry weight (mg shoot$^{-1}$ year$^{-1}$). This method is considered an accurate way of determining long-term annual productivity of seagrasses in the *Posidonia* genus (*Pergent, Pergent-Martini & Cambridge, 1997*; *Peirano, 2002*). As our study is concerned with long-term productivity, this method was used in favour of leaf-marking, which provides a snapshot of leaf productivity at the time of sampling (*Short & Duarte, 2001*). We followed the methods of *Pergent et al. (2004)*, by detaching and numbering the dead leaf sheaths from oldest to youngest and measuring cross sections under the microscope to record the distinct cyclical variations in width (one cycle = one chronological year). Due to the high number of leaf sheaths present on every shoot, a subset of ten shoots per meadow was used to conduct the lepidochronology measures. Primary productivity (PI) was estimated using the formula $PI = N \times L \times D$. Where $N$ is the number of leaves produced annually (number of sheaths per cycle), $L$ is the mean leaf length and $D$ is the mean leaf density (leaf weight per unit length).

Near-infrared reflectance spectroscopy (NIRS) was used to determine the nitrogen content and total phenols within leaf tissues. NIRS is a rapid and inexpensive technique for analysing the composition of organic tissue (*Foley et al., 1998*), and has been used to effectively predict nitrogen and phenolic content in *Posidonia australis* (*Bain, Vergés & Poore, 2013*) and macroalgae (*Hay et al., 2010*). NIRS works by irradiating a sample and measuring the absorbance over a series of wavelength intervals, resulting in a spectrum that characterises the chemical composition of the sample. Quantifying a specific plant trait then depends on constructing a statistical model that calibrates the spectral properties of a sample with the laboratory derived values determined for the trait of interest (*Foley et al.,*

*1998*); in this case, nitrogen content and total phenols. Samples were processed and spectra were collected following the methods outlined in *Bain, Vergés & Poore (2013)*. Prediction models were developed by calibrating NIRS spectra from a subset of samples against laboratory reference values using WinISI$^{TM}$ III software (Infrasoft International, State College, PA). Reference values for nitrogen content were determined by combustion using a CHN analyser (TruSpec® Micro Series, Michigan). The concentration of total phenols was obtained colorimetrically using the Folin-Ciocalteu's method (*Singleton, Orthofer & Lamuela-Raventós, 1999*; *Ainsworth & Gillespie, 2007*) with a gallic acid standard. Prediction models from *Bain, Vergés & Poore (2013)* for *P. australis* were then used to estimate nitrogen (% dry weight, model $R^2 = 0.95$) and total phenols (gallic acid equivalent (GAE) % dry weight, model $R^2 = 0.92$) in the full set of samples.

Herbivory was measured indirectly on each shoot as the area of leaves missing due to herbivore damage (mm$^2$ shoot$^{-1}$). Missing leaf tips were not included in these calculations as we were unable to distinguish between area lost due to damage sustained during harvesting or storm activity/erosion and true herbivory.

To assess whether differences in tissue loss to herbivory were explained by differences in herbivore abundance among sites (rather than seagrass traits), herbivore abundance was measured within each of the 12 meadows using a combination of three methods: visual transects (*Greene & Alevizon, 1989*), beach seine netting (*Connolly, 1994*), and benthic sleds (*Colman & Segrove, 1955*). Each method was replicated three times per meadow. Visual transects were conducted via snorkel, swimming in a straight line for 50 m (marked with weighted measuring tape). All fish within 2 m on either side of the measuring tape were recorded. A benthic sled was towed by hand along the seagrass canopy at a constant depth of 1 m. The trawls were 100 m in length and were towed for approximately 3.5 min (a speed of about 1 knot depending on prevailing conditions). The trawl frame consisted of a stainless steel sled with an opening frame measuring 80 × 40 cm. This was attached to a conical net measuring 2.4 m in length, with the cod end tapering to 105 mm in diameter (securely fastened during each trawl). Each trawl was hauled by one person at the end of a long tow rope in a line parallel to the shore (see *Colman & Segrove, 1955*). The beach seine net used was 20 m in length (3 m cod end) with a 2 m drop. The net was towed through 1 m of water directly over the seagrass canopy by two people standing parallel to the shore for a distance of 50 m. Nets for both the benthic sled and beach seine were made from 5 mm mesh. All fish caught in the nets were photographed for later identification and returned to the water. Herbivorous fish abundance per meadow was then standardised using catch per unit effort (CPUE).

## Statistical analyses

The coefficient of variation (CV) was used to quantify variation in the phenotypic traits, given that they were measured using different units and had widely different means, where the coefficient of variation, CV, is the standard deviation divided by the mean for each trait within each meadow. This standardised level of variation across all traits allowed contrasts to be made between phenotypic diversity (mean CV across all traits) and genotypic richness (number of MLGs).
Linear regression was used to determine whether genotypic richness could predict the variation in phenotypic traits (as measured by the CV for surface area, shoot biomass, productivity, phenolics, nitrogen, herbivory and epiphyte biomass). For each trait, the mean CV ($\pm$SE) across shoots within a meadow was used. Linear regression analyses were also used to determine potential relationships between genotypic richness and average values for the aforementioned phenotypic traits. To ensure that a linear relationship was the most appropriate fit for the data, we used Akaike's information criterion to compare the best-fit of linear, polynomial and logarithmic regressions.

It is important to note that that an interdependency may exist between genotypic and allelic richness when attempting to estimate the importance of genetic diversity, particularly at low genotypic richness levels (*Massa et al., 2013*). As such, we ran additional linear regression analyses to explore potential relationships between allelic richness per meadow and phenotypic trait means and variation.

To investigate the proportion of phenotypic variation explained by genotypic identity and the environment, we compared (i) variation within and among replicate genotypes in individual meadows, and (ii) variation within and among meadows in MLGs occurring across more than one location. Although we found seven MLGs that were shared across two or more sites (Table 1), only two of these MLGs ('Genotype A' and 'Genotype C') had enough replicates both within and across meadows to compare this variation with sufficient statistical power. We used a series of one-way ANOVAs to compare variation within Genotypes 'A' and 'C' across the two meadows in which each occurred (Genotype A in Lake Macquarie/Wallis Lake, and Genotype C in Port Stephens/Pittwater) to determine the amount of phenotypic variation explained by environment (meadow). As genotypes 'A' and 'C' were not found in the same locations, a factorial ANOVA testing for the effect of genotype, environment and genotype by environment interactions was not possible. The meadows at Port Stephens and Pittwater each contained enough individual replicated genotypes to compare how phenotypic traits varied within and among different MLGs in the same meadow. In this way we could determine the amount of phenotypic variation explained by genotype not confounded by differences in the environment. Again, separate analyses had to be used for the two meadows as these genotypes were not shared across locations.

## RESULTS

Positive linear relationships were found between the number of genotypes per meadow and the mean of the coefficient of variation for four traits (surface area, biomass, productivity and epiphyte load). Considering each trait alone, the strongest positive linear relationship was between genotypic richness and variation in surface area (Fig. 2A). Similarly, greater variation in biomass per shoot was significantly related to high genotypic richness (Fig. 2B). There were no significant relationships between genotypic richness and variation in nitrogen content or total phenols (Fig. 2C). The relationship between mean shoot density and genotypic richness was also not significant (Fig. S1).

Meadows with greater genotypic richness did not show significantly greater productivity per shoot on average (Fig. S1). However, there was a positive linear relationship between

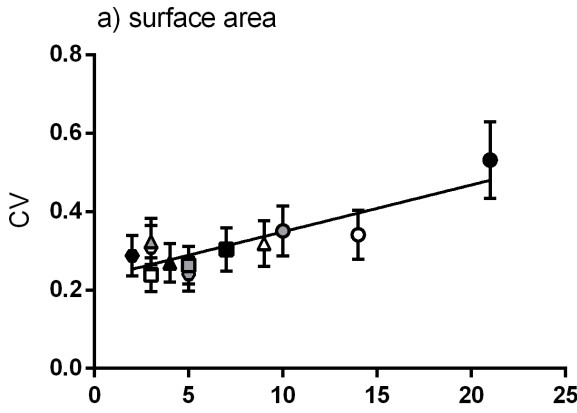

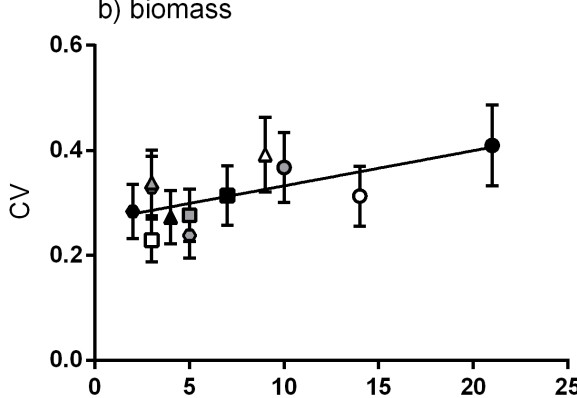

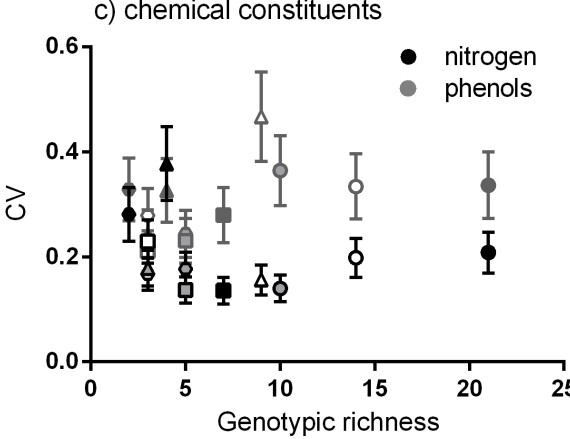

**Figure 2 Relationship between genotypic richness and variation in surface area, biomass and chemical constituents.** The relationship between genotypic richness and phenotypic dissimilarity (measured as the coefficient of variation or CV) for (A) surface area (mm$^2$ shoot$^{-1}$); (B) biomass (mg dw shoot$^{-1}$); and (C) chemical constituents (GAE % dw shoot$^{-1}$). Significant relationships from linear regression were found between genotypic richness and the coefficients of variation for surface area (A; $R^2 = 0.75$, $P < 0.001$) and shoot biomass (B; $R^2 = 0.43$, $P = 0.02$). There were no significant relationships between genotypic richness and the coefficients of variation for nitrogen or phenols (C; $R^2 = 0.04$, $P = 0.56$; $R^2 = 0.20$, $P = 0.15$, respectively). Different symbols correspond to individual meadows sampled (see Fig. 1).
genotypic richness and *variation* in shoot productivity (Fig. 3A), denoting a greater range of productivity values in the meadows with more genotypes. To test whether the meadows with greatest numbers of genotypes showed greater capacity for high shoot productivity, we performed a quantile regression on the raw productivity data (Fig. S2). We detected no significant increase in the highest levels of shoot productivity (values in the 90th percentile) with greater genotypic richness, but did detect a negative relationship between genotypic richness and the likelihood of having low productivity shoots (the 10th percentile; Fig. S2).

Of the community traits measured, a significant linear relationship was found between genotypic richness and variation in epiphyte biomass (Fig. 3B). Genotypic richness and mean herbivory levels showed no significant linear relationship (Fig. 3C). While there were no significant non-linear relationships found either, there is a clear increase in mean herbivory with genotypic richness up to 14 genotypes, after which there is a drop in mean herbivory to near zero at the highest level of diversity (21 genotypes, Fig. 3C). This is not a direct result of the number of herbivores per meadow (rather than genotypes) as the relationship between herbivore abundance (CPUE) and mean herbivory (Fig. S3) does not match that between herbivory and genotypic richness (Fig. 3C). This relationship was also tested using the diversity (Shannon diversity index) of herbivorous fish and mean herbivory, and the result was also non-significant ($R^2 = 0.01$, $P = 0.78$). As epiphytic biomass may be related to levels of herbivory (*Heck & Valentine, 2006*), the relationship between these variables was also explored. However, there was no relationship between tissue loss to herbivory and epiphyte load ($R^2 = 0.14$, $P = 0.24$), nor was there a relationship between the abundance of herbivorous fish and epiphyte load ($R^2 = 0.05$, $P = 0.47$).

There were no significant linear relationships between the mean values of each phenotypic trait measured and genotypic richness. That is, despite significant relationships between genotypic richness and variation in some plant traits, shoots were not significantly larger on average in meadows with more genotypes, nor did they grow more densely (Fig. S4).

Additional linear regression analyses were conducted to explore potential relationships between allelic richness per meadow and phenotypic trait means and variation. There were no significant relationships observed between allelic richness and variation in any of the traits measured. Similarly, there were no significant relationships observed between allelic richness and trait means, with the exception of mean tissue lost to herbivory, which significantly increased with increasing allelic richness ($R^2 = 0.39$; $P = 0.03$; Fig. S5).

Environment (i.e., meadow) explained between 16–33% of phenotypic variation in the two MLGs that occurred across two geographically distinct meadows (Table 2; 'Genotype A' occurring in both Wallis Lake and Lake Macquarie, and 'Genotype C' occurring in both Port Stephens and Pittwater; Table 1). For Genotype A, there were significant differences between meadows in surface area, biomass, leaf nitrogen content and epiphyte load across the two locations. For Genotype C, the results revealed no significant differences in any of the phenotypic traits measured for this genotype (Table 2A), with the exception of biomass, which was significantly lower in Pittwater than in Port Stephens ($F_{1,20} = 9.51$, $P = 0.005$).

When contrasting genotypes within the two meadows with multiple, replicated genotypes (Port Stephens and Pittwater), we found no significant differences among genotypes,

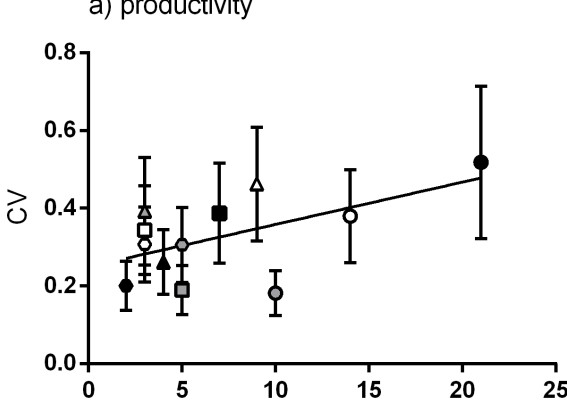

a) productivity

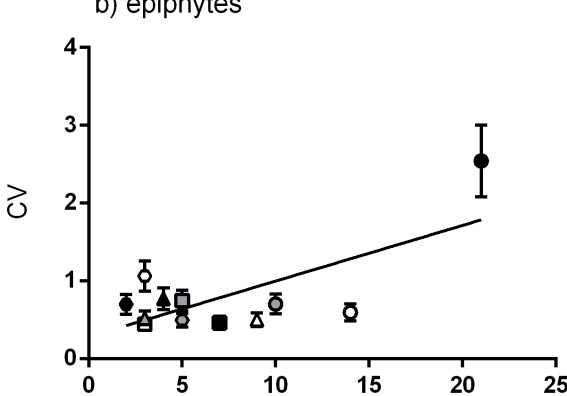

b) epiphytes

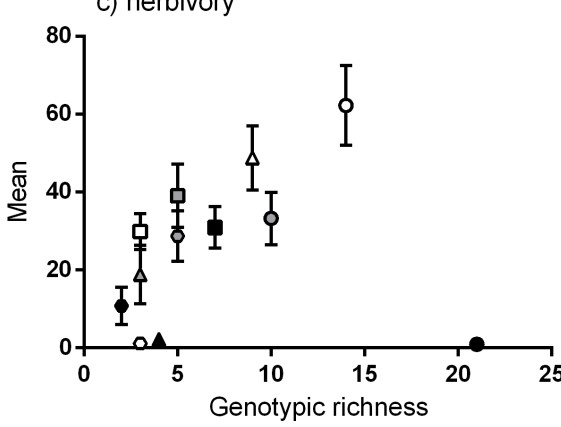

c) herbivory

**Figure 3 Relationships between genotypic richness and variation in productivity, variation in epiphyte biomass and mean herbivory.** The relationships between genotypic richness and (A) mean coefficient of variation for productivity (mg dw shoot$^{-1}$ year$^{-1}$); (B) mean coefficient of variation for epiphyte biomass (mg dw shoot$^{-1}$); and (C) mean herbivory (mm$^2$ shoot$^{-1}$). There was a significant relationship between genotypic richness and mean coefficient of variation for both productivity (B; $R^2 = 0.32$, $P = 0.05$) and epiphyte biomass (A; $R^2 = 0.48$, $P = 0.02$). There was no significant linear relationship between mean herbivory and genotypic richness ($R^2 = 0.03$, $P = 0.6$). Different symbols correspond to individual meadows sampled (see Fig. 1).
**Table 2 Percentage of variation attributable to environment and genotype.** The percentage of variation attributable to environment (meadow) and genotype for (A) the genotypes (A and C) that were shared between locations, and (B) for the two locations in which several replicated genotypes occurred (four within Port Stephens and two within Pittwater). For each trait, the percentage of variation explained by the factor and the residual (other) was calculated from least squares variance components following one way ANOVA. Significant effect of the ANOVAs (*P*-values <0.05) are in bold. Asterisks represent missing data.

**(A)**

| Plant trait | Genotype A | | Genotype C | |
|---|---|---|---|---|
| | Environment | Other | Environment | Other |
| Surface area | **30.0%** | **70.0%** | 17.6% | 82.4% |
| Biomass | **16.0%** | **84.0%** | **33.4%** | **66.6%** |
| Nitrogen | **31.7%** | **68.4%** | 0.9% | 99.1% |
| Phenols | 6.7% | 93.4% | 1.1% | 99.0% |
| Productivity | 17.3% | 82.7% | 18.7% | 81.3% |
| Herbivory | 5.1% | 94.9% | 4.9% | 95.1% |
| Epiphytes | **31.5%** | **68.5%** | 15.4% | 84.6% |

**(B)**

| Plant trait | Port Stephens | | Pittwater | |
|---|---|---|---|---|
| | Genotype | Other | Genotype | Other |
| Surface area | 27.2% | 72.8% | 9.7% | 90.3% |
| Biomass | 23.8% | 76.2% | 6.5% | 93.5% |
| Nitrogen | 10.1% | 89.9% | 15.0% | 85.0% |
| Phenols | **38.9%** | **61.1%** | 0.1% | 99.9% |
| Productivity | * | * | * | * |
| Herbivory | 5.40% | 94.6% | 0.5% | 99.5% |
| Epiphytes | 26.7% | 73.3% | 1.3% | 98.7% |

except for total phenolics in Port Stephens (Table 2B; $F_{3,22} = 4.67$, $P = 0.011$). In that case, approximately 39% of the variation observed was explained by genotype.

## DISCUSSION

Our results clearly demonstrate that intraspecific genotypic diversity is a strong predictor of phenotypic variation in multiple functionally important traits of the endangered clonal seagrass, *Posidonia australis*. Genotypic richness most strongly predicted variation in morphological traits (surface area and biomass), as well as variation in shoot productivity of individual seagrass meadows.

Genetic variation should only be advantageous to a population if it translates into phenotypic variation; the raw material for evolution by natural selection (*Foster Huenneke, 1991*). Although the microsatellite loci used in this study are assumed to come from non-coding (neutral) regions of the genome, our results suggest that these genetic markers adequately reflect variation in genes coding for quantitative traits, such as leaf morphology and productivity. This variation should be especially important for predominantly clonal species like *P. australis* because it is highly unlikely that a single clone would be able to capitalise on all levels of performance. Rather, it is more likely that individual genotypes that do not contribute to one ecosystem process may play an important role in a separate process and/or under different conditions (as observed in studies of species diversity; (*Duffy, Richardson & Canuel, 2003*; *Bradford et al., 2014*; *Byrnes et al., 2014*; *Lefcheck et al., 2015*)). Correspondingly, our results suggest that a variety of genotypes and corresponding phenotypes is the best case scenario for adaptive capacity in terms of both responding to change and recovering from disturbance (*Hughes & Stachowicz, 2004*; *Reusch et al., 2005*; *Ehlers, Worm & Reusch, 2008*; *Hughes & Stachowicz, 2011*; *Hughes, 2014*).

Our original hypothesis that average productivity would be higher in more diverse meadows was based on results from a number of similar studies linking genotypic diversity to plant production (*Hughes & Stachowicz, 2004*; *Reusch et al., 2005*;

*Crutsinger et al., 2006*; *Kotowska, Cahill & Keddie, 2010*; *Hughes & Stachowicz, 2011*; *Drummond & Velland, 2012*). However, we did not find a clear relationship between mean shoot productivity and genotypic richness. Interestingly, we found that the variation of productivity values was significantly greater in more genotypically diverse meadows (Fig. 3A). While productivity was more variable, we did not find that the capacity for high shoot productivity was any greater in meadows with more genotypes (i.e., the highest values of productivity in these meadows were no higher than those in meadows with few genotypes). These results may be reflecting a sampling effect (*Wardle, 1999*), meaning that there is a higher probability of selecting a wider variety of phenotypes from a more diverse group (*Hughes et al., 2008*; *Forsman & Wennersten, 2015*). As such, we may expect that, on average, populations with a higher genetic diversity may be more likely to include genotypes with a wider range of productivity values (both higher and lower than the expected average).

The relationship between genotypic richness and tissue loss to herbivory appeared to be non-linear, with the greatest amount of herbivore damage occurring at intermediate levels of genotypic richness, after which there is a marked drop in herbivory at the meadow of highest diversity (St. Georges Basin; 21 MLGs). This drop in herbivory is consistent with studies that report a greater resistance to herbivory in more genotypically diverse plant communities (*Hughes & Stachowicz, 2004*; *McArt & Thaler, 2013*). While we did not find any significant relationships between genotypic richness and herbivory, we did find a positive linear relationship between allelic richness and herbivory. This discrepancy between diversity measures can be attributed to the meadow at St. Georges Basin, which had the highest genotypic richness of all meadows sampled, but had intermediate allelic richness (15 alleles per meadow), compared to other locations which had up to 20 alleles per meadow (Pambula Lake, Sydney Harbour and Port Hacking). The interdependent relationship between allelic richness and genotypic richness is considered important in the context of resilience and adaptive capacity of seagrasses (*Massa et al., 2013*; *Jahnke, Olsen & Procaccini, 2015*), and the effects of these two sources of genetic diversity are often hard to disentangle at low levels of genotypic richness (*Massa et al., 2013*). Overall, our results suggest that an increase in genetic diversity is related to increased tissue loss to herbivory. Similar findings of increased host plant genetic diversity influencing levels of herbivore damage have been reported for experimental manipulations of terrestrial plants (e.g., *Castagneyrol et al., 2012*; *Barton et al., 2015*). In these examples, it was concluded that complementarity among genotypes, rather than the selection effect, was the mechanism behind increased insect herbivory in genetically diverse plots, with assemblages of different genotypes benefiting polyphagous herbivores (*Castagneyrol et al., 2012*), and different feeding guilds (*Barton et al., 2015*). In both cases, genetic diversity was a poor predictor of herbivore abundance.

It is commonly observed that epiphytic biomass may be related to herbivory in seagrass beds (*Heck & Valentine, 2006*), but our data showed that neither the amount of leaf tissue lost to herbivory, nor the number of herbivorous fish were related to epiphyte load. A positive relationship was found between genotypic richness and variation in epiphyte load, however, this relationship appears to be strongly influenced by a single site (St. Georges
Basin). This relationship should therefore be interpreted with caution, particularly given the high temporal variability in epiphytic growth at the locations sampled.

Seagrasses of the genus *Posidonia* are considered phylogenetically conservative, with low resolution among the eight Australian species across multiple chloroplast and nuclear coding gene regions (*Aires et al., 2011*). Despite this, seagrasses within this group are thought to have existed for more than 60 million years (*Aires et al., 2011*) in highly variable physical conditions around the temperate Australian coastline. The *P. australis* meadows on the east coast of Australia are presumably long-established (since sea levels stabilised ~6,500 years ago following the last glacial maximum), but are highly fragmented and hence unlikely to experience any contemporary gene flow via pollen or seed dispersal (*Evans et al., 2014*). The clones existing within these locations are thus assumed to be highly plastic and capable of withstanding local environmental changes. Although our study shows that phenotypic variation is greatest in the most genotypically diverse meadows, we also detected some phenotypic variation in meadows that are nearly monoclonal (meadows dominated by a single MLG; Wallis Lake, Lake Macquarie, Brisbane Water and Jervis Bay), highlighting substantial plasticity within these genotypes.

Although phenotypic plasticity is considered a strong predictor of fitness and competitive ability (*Callaway, Pennings & Richards, 2003*; *Miner et al., 2005*; *Stomp et al., 2008*), the rate of environmental change experienced by seagrass meadows in the 21st Century is unprecedented (*Unsworth, Van Keulen & Coles, 2014*), and there are ecological limits to phenotypic plasticity that can impact its adaptive value (*Valladares, Gianoli & Gómez, 2007*). Moreover, a plastic response to environmental change (e.g., reduced leaf area under low light conditions) may not actually enhance plant fitness and thus may not necessarily be an adaptive response (*Schlichting, 1986*). Predicted rapid environmental change can therefore potentially put vulnerable clonal plants at risk of extinction (*Honnay & Bossuyt, 2005*; *Jump, Marchant & Peñuelas, 2009*), and populations with a greater variety of genotypes and corresponding phenotypes would be considered less at risk than populations with low diversity, regardless of plasticity (*Forsman & Wennersten, 2015*).

With few genotypes shared across sampling locations, we had a limited ability to quantify the relative importance of genotype and local environmental conditions on phenotypic traits. We could, however, contrast two abundant MLGs ('Genotype A' and 'Genotype C') that were each shared across two meadows (Wallis Lake/Lake Macquarie, and Port Stephens/Pittwater respectively, which in both cases are separated by more than 150 km). Given the seed dispersal ecology of *P. australis*, it is considered extremely unlikely that these meadows would experience contemporary gene flow at this distance (*Kendrick et al., 2012*; *McMahon et al., 2014*). As such, we can be confident that these genotypic clones are most likely to have arisen from common ancestral meadows. While only eight microsatellite loci were used to identify genotypic clones in this study, all loci were highly polymorphic with sufficient power to distinguish individual genotypes ($P_{gen} < 0.02$). Thus, it is highly unlikely that shared genotypes across geographically distinct meadows are the result of low marker resolution. While we cannot confirm that the full genomes are an exact match without further analyses, we can be certain that these meadows are at the very least, highly related. Our results showed that the highly dominant MLG occurring in both Wallis Lake

and Lake Macquarie ('Genotype A') significantly varied in a number of phenotypic traits across the two geographically distinct locations (including leaf surface area, biomass and nitrogen content). Between 16 and 32% of this variation was explained by 'environment' (meadow), suggesting that this genotype is capable of remarkable plasticity.

In contrast, the dominant MLG occurring in both Port Stephens and Pittwater ('Genotype C') showed no significant differences across the two meadows for five of the six phenotypic traits measured, suggesting that either this genotype has a lower capacity for phenotypic plasticity, or that the environmental conditions are broadly similar in these two estuaries. This second option is unlikely, given that Pittwater is a highly urbanised meadow with an endangered status and the seagrass meadows are considered in rapid decline (*Creese et al., 2009*; *NSW Department of Primary Industries, 2012*). Biomass was significantly lower in Pittwater when compared to Port Stephens (despite no significant differences across all different genotypes sampled within each location). Surface area of the leaves for 'Genotype C' was no different in these two locations, so it could be hypothesised that lower leaf biomass in Pittwater results from thinner/structurally weaker leaves, perhaps as a consequence of stress caused by reduced light availability (*Ralph et al., 2007*).

When contrasting genotypes within the two meadows that had several replicate genotypes (Pittwater and Port Stephens), there was no detectable variation in phenotypic traits among genotypes (except phenols at Port Stephens). This suggests that these genotypes express largely similar phenotypes within a meadow. Without transplanting these individual genotypes to other environments, it is not possible to determine whether this consistency is due to these genotypes having a similar fixed expression of traits, or all having sufficient phenotypic plasticity to express the same trait values in that environment.

## Conclusion

Our results indicate that phenotypic diversity in populations of *Posidonia australis* increases significantly with increased genotypic diversity (MLG). This suggests that the use of presumably neutral DNA markers to measure genetic diversity also adequately reflects variation in several selectively relevant genes coding for quantitative traits, such as leaf morphology and productivity. Despite evidence of remarkable plasticity within genotypes of *P. australis*, an increase in the variation of functionally relevant traits is expected to be advantageous in the face of environmental change.

## ACKNOWLEDGEMENTS

The genotyping was undertaken at the Botanic Gardens and Parks Authority in Perth. We thank G Truong for lepidochronology data, and T Peters, R Neumann, M Garcia-Piza and R Blick for assistance with fieldwork. AR Hughes and JT Wright provided comments that significantly improved the manuscript.

### Funding
Funding was provided by an Australian Postgraduate Award and an Australian Research Council linkage grant (LP100200429 and LP130100918) with industry partners Cockburn Cement, Department of Parks and Wildlife Western Australia, Botanic Gardens and Parks Authority Western Australia. The funders had no role in study design, data collection and analysis, decision to publish, or preparation of the manuscript.

### Grant Disclosures
The following grant information was disclosed by the authors:
Australian Postgraduate Award.
Australian Research Council linkage grant: LP100200429, LP130100918.

### Competing Interests
The authors declare there are no competing interests.

### Author Contributions
- Suzanna M. Evans conceived and designed the experiments, performed the experiments, analyzed the data, wrote the paper, prepared figures and/or tables, reviewed drafts of the paper.
- Elizabeth A. Sinclair performed the experiments, contributed reagents/materials/analysis tools, reviewed drafts of the paper.
- Alistair G B. Poore and Adriana Vergés conceived and designed the experiments, reviewed drafts of the paper.
- Keryn F. Bain performed the experiments, analyzed the data, contributed reagents/materials/analysis tools, reviewed drafts of the paper.

### Field Study Permissions
The following information was supplied relating to field study approvals (i.e., approving body and any reference numbers):
NSW Government Department of Primary Industries Scientific Collection Permit P11/0059-1.2.

### Data Availability
The raw data was supplied as Data S1.

### Supplemental Information
Supplemental information for this article can be found online at http://dx.doi.org/10.7717/peerj.1633#supplemental-information.

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
