# Peer review of "Genotypic richness predicts phenotypic variation in an endangered clonal plant"

_PeerJ, doi:10.7717/peerj.1633_

## Round 0.1 · original submission · Major Revisions

Both referees are positive about the value of your manuscript, but both also have suggestions to improve the current submission. The first referee has some suggestions for minor revisions that should be easily incorporated. The second referee has some more substantial comments, in particular asking if there is a mistake in the figures and posing the question of whether or not allelic richness contributes to understanding of the results. It again seems to me relatively simple to add the requested genotyping statistics to the manuscript. However, I also agree with this referee that your manuscript would benefit from a power analysis to highlight your confidence in the data. Although it seems to me a relatively simple addition that would prevent future readers from asking the same questions, the addition of a new analysis to the paper classifies as a major revision in my view. Having said that, however, I see nothing in the reviews to suggest that the paper cannot be easily revised as suggested, and I look forward to seeing your revised manuscript. Depending on your detailed response to the referee comments, I will determine at that time whether or not the paper needs to go back to the referees for additional feedback.

·

Basic reporting

In several places in the introduction, a single citation is provided for a broad statement (e.g., lines 46-48, 67-70). I appreciate that the authors have tended to include earlier references rather than relying solely on the most recent literature, but some more contemporary references would help place this work within the broader field of knowledge.

Several papers have promoted the importance of multifunctionality to studies of biodiversity (e.g., Byrnes et al. 2014 Methods in Ecology and Evolution; Bradford et al. 2014 PNAS; and see earlier arguments in Duffy et al. 2003). This concept and literature are relevant for the argument that no single clone can maximize all aspects of performance and should be cited.

The discussion of the sampling effect (lines 337-346) is unclear and would benefit from additional references in support of the hypothesized mechanisms.

Lines 88-90 – Tomas et al. 2011 examined the effects of clonal identity, not clonal diversity

Line 92 – Hughes and Stachowicz 2011 is a more appropriate reference than Hughes et al. 2009 for this statement

Lines 378-381 – citation in support of this statement?

Experimental design

The questions and methods were clearly described.

Validity of the findings

Question 3 examines whether genotypic diversity is related to herbivory or epiphyte load, but these latter 2 (herbivory and epiphyte load) could also be correlated with one another. Was this potential relationship examined?

In general, a multivariate approach that examines how genotypic richness predicts variation in multiple traits (rather than the average across all traits) would be more appropriate to help account for potential co-variation in the traits measured.

An outlier site (St. Georges Basin) was removed from the epiphyte analysis, but was this also done for the other response variables? That site appears to be a major contributor to all of the significant patterns, particularly tissue loss to herbivory. Some statistical justification for the decision to include or exclude that point is needed.

I appreciate the assessment of variation within the genotypes that spanned multiple sites, as well as within and among the genotypes with sufficient replication within a single site. However, I am unclear on the specifics of the analyses. For the genotypes spanning multiple sites, did the analysis include an effect of genotype, an effect of environment, as well as their potential interaction? For the genotypes within the same site, what was included in the model?

Please be sure to clearly differentiate between per shoot productivity (e.g., line 266) and meadow productivity (line 274) throughout the ms.

Can you provide some discussion of the apparent discrepancy between the presence of MLGs occurring at sites separated by greater than 150km (lines 382-391) and the assertion that meadows are unlikely to experience contemporary gene flow? (lines 364-365)

The concluding sentence is at odds with your own data, in that MLG diversity was not clearly related to meadow productivity. Consider rewording.

Additional comments

Evans et al. present the results of a study examining the relationships among genotypic richness and phenotypic variation in natural populations of the seagrass Posidonia australis. I have reviewed a previous version of this manuscript, and the authors have addressed many of my prior comments. I have a few additional suggestions that I think can be readily addressed and will improve the clarity and context of the manuscript.

Reviewer 2 ·

Basic reporting

The goal of the study (to investigate the relationship between genotypic richness and phenotypic variation in Posidonia) is clearly defined and the importance/relevance of this topic is well placed in the broader context. Regarding the figures, some questions raised below (see ‘Validity of the Findings’) may be addressed with a slight modification of the figures; consider using a different symbol for each meadow throughout the figures to make it easier to relate and compare variables within meadows (i.e., filled square for Wallis Lake, open circle for Port Stephens, etc.). In addition, what are the units of mean herbivory (Fig. 4c)?

Experimental design

The manuscript clearly defines the research questions (though consider rephrasing question 3 (line 132) to predict the relationship between genotypic diversity and herbivory/epiphyte load). The methods are well described and appropriate for the questions being addressed (but see ‘Validity of the Findings’ regarding a couple of methodological details related to the analysis).

Validity of the findings

The comparison of the MLGs that occurred in geographically distinct meadows is interesting, but what is the likelihood that genotype A in Wallis Lake and genotype A in Lake Macquarie are actually the same genotype (especially given the fact that genotype A is not present in Port Stephens, which separates these meadows)? The same is true for genotype C (e.g., present in Port Stephens and Pittwater, but not in Lake Macquarie and Brisbane Water, which separate these meadows). It is possible that 8 microsatellite loci may be sufficient to differentiate genotypes in this system, but it’s difficult to judge given little information on the genotyping statistics (e.g., allele number, heterozygosity, etc.). While some of this information is presented in Evans et al. 2014, a general statement regarding the authors’ power to differentiate genotypes based on the 8 loci is needed to justify the claim that A in Wallis Lake and A in Lake Macquarie are likely identical (and similarly C in Port Stephens and C in Pittwater). It is difficult to judge the feasibility and accuracy of the related analysis comparing A and C in different meadows as a result (e.g., does genotype A vary in phenotype across sites because it’s actually genotype A1 and genotype A2?).

In lines 265-267, the traits used to calculate the mean coefficient of variation (Fig. 2) are described as being “directly related to the shoot itself”; however, shoot density (as described in the Methods) is not a shoot-specific trait and thus should likely not be included in calculation of mean CV for shoot morphology (nor in the analysis quantifying the importance of genotype versus environment on phenotype).

In the herbivory results, there looks to be a mistake in either Fig. 4c or Fig. S3: the y-axis is identical (i.e., mean herbivory) in both figures, yet the range of values is different (0 – 60 in Fig. 4c versus 0-40 in Fig. S3). In addition, while the authors recognize that it’s not possible to identify the cause(s) of the low mean herbivory in the high genotypic richness treatment (based on one data point), they suggest that this is not simply a result of the number of herbivores in the meadow. But it’s difficult to determine from the supplemental figure what herbivore CPUE is for this site; to make the discussion of herbivory, herbivore abundance, and genotypic richness more informative, consider using a unique symbol to identify each meadow in all of the figures to make comparisons within meadows possible.

Additional comments

The paper would benefit from consideration of allelic richness (in addition to genotypic richness) as a measure of genetic diversity. Allelic richness may help inform some of the unresolved relationships and it would be interesting to know whether a similar pattern is observed with a different metric of diversity. Because the authors have the data to address this question, it at least merits mention in the discussion to place this study in the context of genetic diversity studies exploring different metrics of diversity.

Similarly, it would be worth examining the relationship between CV (either of all traits or of individual traits) and mean productivity, mean shoot density, etc., given that there was no relationship between genotypic richness and productivity or density. This is a missed opportunity to link phenotypic trait variation (which the authors nicely justify as being of importance to ecosystem productivity and resiliency) to ecosystem function. There may be no relationship (it’s difficult to envision the result of combining multiple figures), but it’s worth investigating.

---

## Round 0.2 · Minor Revisions

Both referees agree that your revisions have satisfied their concerns and that the resubmission is greatly improved. There are a couple minor typos that remain and the referees have a couple additional suggestions for potential improvement to the manuscript. At this point, these are relatively minor, so I leave it to the authors whether they wish to incorporate these suggestions. I am returning the manuscript for you to look at these comments and correct the typos before final acceptance and moving the manuscript forward into production.

·

Basic reporting

Evans et al. present the results of a study examining the relationships among genotypic richness and phenotypic variation in natural populations of the seagrass Posidonia australis. The authors have thoroughly and effectively addressed my prior suggestions, and I think the manuscript will make a valuable and important contribution to the literature.

Experimental design

No comments

Validity of the findings

Minor edit - Reviewer 2 raised a very good point regarding the potential power (or lack thereof) of the microsatellite loci to distinguish genotypes. I recommend that the authors calculate and present Pgen (also referred to as Psex) to clarify this issue.

Line 385 – “between allelic richness AND herbivory”

Reviewer 2 ·

Basic reporting

In general, the authors do a nice job of responding to feedback from the reviewers. In particular, the edits suggested by reviewer 1, which have been nicely incorporated by the authors, have greatly helped to place this study and its results in a broader context.

The incorporation of different symbols in the figures to represent the different meadows is also helpful; however, consider modifying the legend for Fig. 2c, as it’s somewhat difficult to determine which symbols represent nitrogen and which represent phenols.

Experimental design

The authors mention in their response to reviewer 1’s suggestion to use a multivariate approach (comment #9) that some variables were not measured for every individual. While this is understandable, the text indicates that “all shoots collected for genotyping were measured for traits relating directly to the shoot phenotype…” (lines 187-188). For clarity and for help in understanding the statistical approach used to analyze the data, it would be useful to report what subset of the shoots were measured for each trait (or similar) to clarify the methods used in this study. In addition, a quick mention of whether assumptions of normality, etc. were tested prior to analysis (and if any data transformations were necessary) would be worthwhile.

Validity of the findings

It would be helpful to know the relationship between genotypic richness and allelic richness in interpreting the results – was there a strong correlation between these metrics? (This information would nicely complement the discussion of differences in relative genotypic and allelic richness in St. Georges Basin in lines 414-418).

The authors’ response to both reviewers’ questions about the presence of MLGs at sites separated by >150 km despite limited contemporary gene flow is helpful and informative. It would be nice to include a brief reference to some of these points in the text, clarifying why they think these clones originated from a common ancestral meadow and acknowledging that they may not be exact clones (but are highly related).

---

## Round 0.3 · accepted · Accept

The authors have addressed the primary concerns of both referees and I am content with the response on the areas of disagreement. I am happy to move this manuscript forward.